# DIFFUSION MODULATION VIA ENVIRONMENT MECHANISM MODELING FOR PLANNING

## ABSTRACT

Diffusion models have shown promising capabilities in trajectory generation for planning in offline reinforcement learning (RL). However, conventional diffusion-based planning methods often fail to account for the fact that generating trajectories in RL requires unique consistency between transitions to ensure coherence in real environments. This oversight can result in considerable discrepancies between the generated trajectories and the underlying mechanisms of a real environment. To address this problem, we propose a novel diffusion-based planning method, termed as Diffusion Modulation via Environment Mechanism Modeling (DMEMM). DMEMM modulates diffusion model training by incorporating key RL environment mechanisms, particularly transition dynamics and reward functions. Experimental results demonstrate that DMEMM achieves state-of-the-art performance for planning with offline reinforcement learning.

## 1 INTRODUCTION

Offline reinforcement learning (RL) has garnered significant attention for its potential to leverage pre-collected datasets to learn effective policies without requiring further interaction with the environment (Levine et al., 2020). One emerging approach within this domain is the use of diffusion models for trajectory generation (Janner et al., 2022). Diffusion models (Sohl-Dickstein et al., 2015; Ho et al., 2020), initially popularized for tasks such as image synthesis, have demonstrated promising capabilities in generating coherent and diverse trajectories for planning in offline RL settings (Janner et al., 2022; Ni et al., 2023; Li, 2024; Goyal and Grand-Clement, 2023). Nevertheless, the essential differences between mechanisms in image synthesis and RL necessitate specific considerations for the effective application of diffusion models in RL.

In image synthesis (Ho et al., 2020), diffusion models primarily aim to produce visually coherent outputs consistent in style and structure, while RL tasks demand environment and task oriented consistency between transitions in the generated trajectories (Janner et al., 2022) to ensure that the generated sequences are not only plausible but also effective for policy learning (Kumar et al., 2020). This consistency is essential for ensuring that the sequence of actions within the generated trajectories can successfully guide the RL agent from the current state to the target state. However, conventional diffusion-based planning methods often overlook this need for transition coherence (Janner et al., 2022). By simply adopting traditional diffusion models like DDPM, which utilize a fixed isotropic variance for Gaussian distributions, such diffusion-based planning models may fail to adequately capture the transition dynamics necessary for effective RL, leading to inaccurate trajectories and suboptimal learned policies (Wu et al., 2019; Niu et al., 2024).

To address this problem, we introduce a novel diffusion-based planning method called Diffusion Modulation via Environment Mechanism Modeling (DMEMM). This method modulates the diffusion process by integrating RL-specific environment mechanisms, particularly transition dynamics and reward functions, directly into the diffusion model training process on offline data, thereby enhancing the diffusion model to better capture the underlying transition and reward structures of the offline data. Specifically, we modify the diffusion loss by weighting it with the cumulative reward, which biases the diffusion model towards high-reward trajectories, and introduce two auxiliary modulation losses based on empirical transition and reward models to regularize the trajectory diffusion process, ensuring that the generated trajectories are not only plausible but also reward-optimized. Additionally, we also utilize the transition and reward models to guide the sampling process dur-

ing planning trajectory generation from the learned diffusion model, further aligning the outputs with the desired transition dynamics and reward structures. We conducted experiments on multiple RL environments. Experimental results indicate that our proposed method achieves state-of-the-art performance compared to previous diffusion-based planning approaches.

This work presents a significant step forward in the application of diffusion models for trajectory generation in offline RL. The main contributions can be summarized as follows:

- We identify a critical problem in conventional diffusion model training for offline RL planning, where fixed isotropic variance and disregard for rewards may lead to a mismatch between generated trajectories and those desirable for RL. To address this, we propose a novel method called Diffusion Modulation via Environment Mechanism Modeling (DMEMM).

- We incorporate RL-specific environment mechanisms, including transition dynamics and reward functions, into diffusion model training through loss modulation, enhancing the quality and consistency of the generated trajectories in a principled manner and providing a fundamental framework for adapting diffusion models to offline RL tasks.

- Our results on multiple RL environments show that the proposed method achieves state-of-the-art performance in offline RL planning, validating the effectiveness of our approach.

## 2 RELATED WORKS

### 2.1 OFFLINE REINFORCEMENT LEARNING

Offline reinforcement learning (RL) has gained significant traction in recent years, with various approaches proposed to address the challenges of learning from static datasets without online environment interactions. Fujimoto et al. (2019) introduced Batch Constrained Q-Learning (BCQ) that learns a perturbation model to constrain the policy to stay close to the data distribution, mitigating the distributional shift issue. Wu et al. (2019) conducted Behavior Regularized Offline Reinforcement Learning (BRAC) that incorporates behavior regularization into actor-critic methods to prevent the policy from deviating too far from the data distribution. Conservative Q-Learning (CQL) by Kumar et al. (2020) uses a conservative Q-function to underestimate out-of-distribution actions, preventing the policy from exploring unseen state-action regions. Kostrikov et al. (2022) conducted Implicit Q-Learning (IQL) to directly optimize the policy to match the expected Q-values under the data distribution. Goyal and Grand-Clement (2023) introduce Robust MDPs to formulate offline RL as a robust optimization problem over the uncertainty in the dynamics model. Planning has emerged as a powerful tool for solving offline RL tasks. MOPO (Yu et al., 2020) incorporates uncertainty-aware planning into offline RL by penalizing simulated trajectories that deviate from the offline dataset.

### 2.2 DIFFUSION MODEL IN REINFORCEMENT LEARNING

Diffusion models have emerged as a powerful tool for RL tasks, particularly in the areas of planning and policy optimization. Janner et al. (2022) first introduced the idea of using diffusion models for trajectory planning in offline RL, casting it as a probabilistic model that iteratively refines trajectories. Li (2024) propose the Latent Diffuser, which generates actions in a latent space by leveraging a Score-based Diffusion Model (SDM) (Song et al., 2021) and employs energy-based sampling to enhance the overall performance of diffusion-based planning. Subsequent work by Venkatraman et al. (2024) further extends this idea and introduces Latent Diffusion-Constrained Q-learning (LDCQ), which learns latent skills and enables the agent to estimate Q-functions directly in the latent space. Ajay et al. (2023) propose Decision Diffuser (DD), a method that leverages classifier-free guidance with low-temperature sampling to condition on returns, constraints, and skills, enabling the generation of high-quality decision-making sequences. Chen et al. (2024) propose a Hierarchical Diffuser, which achieves hierarchical planning by breaking down planning trajectories into segments and treating intermediate states as subgoals to ensure more precise planning. More recently, Ni et al. (2023) proposed a task-oriented conditioned diffusion planner (MetaDiffuser) for offline meta-reinforcement learning. MetaDiffuser learns a context-conditioned diffusion model that can generate task-oriented trajectories for planning across diverse tasks, demonstrating the outstanding conditional generation ability of diffusion architectures. These works highlight the versatility of diffusion models in addressing RL challenges.

## 3 PRELIMINARIES

Reinforcement learning (RL) (Sutton and Barto, 2018) can be modeled as a Markov Decision Process (MDP) $M = (\mathcal{S}, \mathcal{A}, \mathcal{T}, \mathcal{R}, \gamma)$ in a given environment, where $\mathcal{S}$ denotes the state space, $\mathcal{A}$ corresponds to the action space, $\mathcal{T} : \mathcal{S} \times \mathcal{A} \to \mathcal{S}$ defines the transition dynamics, $\mathcal{R} : \mathcal{S} \times \mathcal{A} \to \mathbb{R}$ represents the reward function, and $\gamma$ is a discount factor. Offline RL aims to train an RL agent from an offline dataset $\mathcal{D}$, consisting of a collection of trajectories $\{\boldsymbol{\tau}_1, \boldsymbol{\tau}_2, \cdots, \boldsymbol{\tau}_i, \cdots\}$, with each trajectory $\boldsymbol{\tau}_i = (s_0^i, a_0^i, r_0^i, s_1^i, a_1^i, r_1^i, \ldots, s_T^i, a_T^i, r_T^i)$ sampled from the underlying MDP in the given environment. In particular, the task of planning in offline RL aims to generate planning trajectories from an initial state $s_0$ by simulating action sequences $\boldsymbol{a}_{0:T}$ and predicting future states $\boldsymbol{s}_{0:T}$ based on those actions. The objective is to learn an optimal plan function such that the cumulative reward can be maximized when executing the plan under the underlying MDP of the given environment.

### 3.1 PLANNING WITH DIFFUSION MODEL

Diffusion probabilistic models, commonly known as "diffusion models" (Sohl-Dickstein et al., 2015; Ho et al., 2020), are a class of generative models that utilize a unique Markov chain framework. When applied to planning in offline RL, the objective is to generate best planning trajectories $\{\boldsymbol{\tau}\}$ by learning a diffusion model on the offline RL dataset $\mathcal{D}$.

**Trajectory Representation**   In the diffusion model applied to RL planning, it is necessary to predict both states and actions. Therefore, the trajectory representation in the model is in an image-like matrix format. In particular, trajectories are represented as two-dimensional arrays (Janner et al., 2022), where each column corresponds to a state-action pair $(s_t, a_t)$ of the trajectory:

$$\boldsymbol{\tau} = \begin{bmatrix} s_0 & s_1 & \cdots & s_T \\ a_0 & a_1 & \cdots & a_T \end{bmatrix}$$

**Trajectory Diffusion**   The diffusion model (Ho et al., 2020) comprises two primary processes: the forward process and the reverse process. The forward process (diffusion process) is a Markov chain characterized by $q(\boldsymbol{\tau}^k|\boldsymbol{\tau}^{k-1})$ that gradually adds Gaussian noise at each time step $k \in \{1, \cdots, K\}$, starting from an initial clean trajectory sample $\boldsymbol{\tau}^0 \sim \mathcal{D}$. The conditional probability is particularly defined as a Gaussian probability density function, such as:

$$q(\boldsymbol{\tau}^k|\boldsymbol{\tau}^{k-1}) := \mathcal{N}(\boldsymbol{\tau}^k; (1 - \beta_k)\boldsymbol{\tau}^{k-1}, \beta_k \mathbf{I}), \tag{1}$$

with $\{\beta_1, \cdots, \beta_K\}$ representing a predefined variance schedule. By introducing $\alpha_k := 1 - \beta_k$ and $\bar{\alpha}_k := \prod_{i=1}^{k} \alpha_i$, one can succinctly express the diffused sample at any time step $k$ as:

$$\boldsymbol{\tau}^k = \sqrt{\bar{\alpha}_k}\boldsymbol{\tau}^0 + \sqrt{1 - \bar{\alpha}_k}\boldsymbol{\epsilon}, \tag{2}$$

where $\boldsymbol{\epsilon} \sim \mathcal{N}(\mathbf{0}, \mathbf{I})$. The reverse diffusion process is an iterative denosing procedure, and can be modeled as a parametric Markov chain characterized by $p_\theta(\boldsymbol{\tau}^{k-1}|\boldsymbol{\tau}^k)$, starting from a Gaussian noise prior $\boldsymbol{\tau}^K \sim \mathcal{N}(\mathbf{0}, \mathbf{I})$, such that:

$$p_\theta(\boldsymbol{\tau}^{k-1}|\boldsymbol{\tau}^k) = \mathcal{N}(\boldsymbol{\tau}^{k-1}; \mu_\theta(\boldsymbol{\tau}^k, k), \sigma_k^2 \mathbf{I}), \tag{3}$$

$$\text{with } \mu_\theta(\boldsymbol{\tau}^k, k) = \frac{1}{\sqrt{\alpha_k}} \left( \boldsymbol{\tau}^k - \frac{1 - \alpha_k}{\sqrt{1 - \bar{\alpha}_k}} \epsilon_\theta(\boldsymbol{\tau}^k, k) \right). \tag{4}$$

**Training**   In the literature, the diffusion model is trained by predicting the additive noise $\epsilon$ (Ho et al., 2020) using the noise network $\epsilon_\theta(\boldsymbol{\tau}^k, k) = \epsilon_\theta(\sqrt{\bar{\alpha}_k}\boldsymbol{\tau}^0 + \sqrt{1 - \bar{\alpha}_k}\boldsymbol{\epsilon}, k)$. The training loss is expressed as the mean squared error between the additive noise $\epsilon$ and the predicted noise $\epsilon_\theta(\boldsymbol{\tau}^k, k)$:

$$L_{\text{diff}} = \mathbb{E}_{k \sim \mathcal{U}(1,K), \boldsymbol{\epsilon} \sim \mathcal{N}(\mathbf{0}, \mathbf{I}), \boldsymbol{\tau}^0 \sim \mathcal{D}} \left\| \boldsymbol{\epsilon} - \epsilon_\theta(\sqrt{\bar{\alpha}_k}\boldsymbol{\tau}^0 + \sqrt{1 - \bar{\alpha}_k}\boldsymbol{\epsilon}, k) \right\|^2 \tag{5}$$

where $\mathcal{U}(1, K)$ denotes a uniform distribution over numbers in $[1, 2, \cdots, K]$. With the trained noise network, the diffusion model can be used to generate RL trajectories for planning through the reverse diffusion process characterized by Eq.(3).

## 4 METHOD

In this section, we present our proposed diffusion approach, Diffusion Modulation via Environment Mechanism Modeling (DMEMM), for planning in offline RL. This method integrates the essential transition and reward mechanisms of reinforcement learning into an innovative modulation-based diffusion learning framework, while maintaining isotropic covariance matrices for the diffusion Gaussian distributions to preserve the benefits of this conventional setup—simplifying model complexity, stabilizing training and enhancing performance. Additionally, the transition and reward mechanisms are further leveraged to guide the planning phase under the trained diffusion model, aiming to generate optimal planning trajectories that align well with both the underlying MDP of the environment and the objectives of RL.

### 4.1 MODULATION OF DIFFUSION TRAINING

In an RL environment, the transition dynamics and reward function are two fundamental components of the underlying MDP. Directly applying conventional diffusion models to offline RL can lead to a mismatch between the generated trajectories and those optimal for the underlying MDP in RL. This is due to the use of isotropic covariance and the disregard for rewards in traditional diffusion models. To tackle this problem, we propose to modulate the diffusion model training by deploying a reward-aware diffusion loss and enforcing auxiliary regularizations on the generated trajectories based on environment transition and reward mechanisms.

Given the offline data $\mathcal{D}$ collected from the RL environment, we first learn a probabilistic transition model $\widehat{\mathcal{T}}(s_t, a_t)$ and a reward function $\widehat{\mathcal{R}}(s_t, a_t)$ from $\mathcal{D}$ as regression functions to predict the next state $s_{t+1}$ and the corresponding reward $r_t$ respectively. These models can serve as estimations of the underlying MDP mechanisms. In order to regularize diffusion model training for generating desirable trajectories, using the learned transition model and reward function, we need to express the output trajectories of the reverse diffusion process in terms of the diffusion model parameters, $\theta$. To this end, we present the following proposition.

**Proposition 1.** *Given the reverse process encoded by Eq.(3) and Eq.(4) in the diffusion model, the output trajectory $\widehat{\boldsymbol{\tau}}^0$ denoised from an intermediate trajectory $\boldsymbol{\tau}^k$ at step $k$ has the following Gaussian distribution:*

$$\widehat{\boldsymbol{\tau}}^0 \sim \mathcal{N}(\widehat{\mu}_\theta(\boldsymbol{\tau}^k, k), \widehat{\sigma}^2 \mathbf{I}), \tag{6}$$

$$where \quad \widehat{\mu}_\theta(\boldsymbol{\tau}^k, k) = \frac{1}{\sqrt{\bar{\alpha}_k}} \boldsymbol{\tau}^k - \sum_{i=1}^{k} \frac{1 - \alpha_i}{\sqrt{(1 - \bar{\alpha}_i) \prod_{j=1}^{i} \alpha_j}} \epsilon_\theta(\boldsymbol{\tau}^i, i). \tag{7}$$

Conveniently, we can use the mean of the Gaussian distribution above directly as the most likely output trajectory, denoted as $\widehat{\boldsymbol{\tau}}^0 = \widehat{\mu}_\theta(\boldsymbol{\tau}^k, k)$. This allows us to express the denoised output trajectory explicitly in terms of the parametric noise network $\epsilon_\theta$, and thus the parameters $\theta$ of the diffusion model. Moreover, by deploying Eq.(2), we can get rid of the latent $\{\boldsymbol{\tau}^1, \cdots, \boldsymbol{\tau}^k\}$ and re-express $\widehat{\boldsymbol{\tau}}^0$ as the following function of a sampled clean trajectory $\boldsymbol{\tau}^0$ and some random noise $\boldsymbol{\epsilon}$:

$$\widehat{\boldsymbol{\tau}}_\theta^0(\boldsymbol{\tau}^0, k, \boldsymbol{\epsilon}) = \boldsymbol{\tau}^0 + \frac{\sqrt{1 - \bar{\alpha}_k}}{\sqrt{\bar{\alpha}_k}} \boldsymbol{\epsilon} - \sum_{i=1}^{k} \frac{1 - \alpha_i}{\sqrt{(1 - \bar{\alpha}_i) \prod_{j=1}^{i} \alpha_j}} \epsilon_\theta \left(\sqrt{\bar{\alpha}_i} \boldsymbol{\tau}^0 + \sqrt{1 - \bar{\alpha}_i} \boldsymbol{\epsilon}, i\right). \tag{8}$$

Next, we leverage this output trajectory function to modulate diffusion model training by developing novel auxiliary modulation losses.

### 4.1.1 TRANSITION-BASED DIFFUSION MODULATION

As previously discussed, the deployment of a fixed isotropic variance in conventional diffusion models has the potential drawback of overlooking the underlying transition mechanisms of the RL environment. As a result, there can be potential mismatches between the transitions of generated trajectories and the underlying transition dynamics. Consequently, the RL agent may diverge from the expected states when executing the planning actions generated by the diffusion model, leading to poor planning performance. To address this problem, the first auxiliary modulation loss is designed

to minimize the discrepancy between the transitions in the generated trajectories from the diffusion model and those predicted by the learned transition model $\widehat{\mathcal{T}}$, which encodes the underlying transition mechanism. Specifically, for each transition $(s_t, a_t, s_{t+1})$ in a generated trajectory $\widehat{\boldsymbol{\tau}}_\theta^0(\boldsymbol{\tau}^0, k, \boldsymbol{\epsilon})$, we minimize the mean squared error between $s_{t+1}$ and the predicted next state using the transition model $\widehat{\mathcal{T}}$. This leads to the following transition-based diffusion modulation loss:

$$L_{\text{tr}} = \mathbb{E}_{k \sim \mathcal{U}(1,K), \boldsymbol{\epsilon} \sim \mathcal{N}(\mathbf{0}, \mathbf{I}), \boldsymbol{\tau}^0 \sim \mathcal{D}} \left[ \sum_{(s_t, a_t, s_{t+1}) \in \widehat{\boldsymbol{\tau}}_\theta^0(\boldsymbol{\tau}^0, k, \boldsymbol{\epsilon})} \left\| s_{t+1} - \widehat{\mathcal{T}}(s_t, a_t) \right\|^2 \right] \qquad (9)$$

Here, the expectation is taken over the uniform sampling of time step $k$ from $[1 : K]$, the random sampling of noise $\boldsymbol{\epsilon}$ from a standard Gaussian distribution, and the random sampling of input trajectories from the offline training data $\mathcal{D}$. Through function $\boldsymbol{\tau}_\theta^0$, this loss $L_{\text{tr}}$ is a function of the diffusion model parameters $\theta$. By minimizing this transition-based modulation loss, we enforce that the generated trajectories from the diffusion model are consistent with the transition dynamics expressed in the offline dataset. This approach enhances the fidelity of the generated trajectories and improves the overall performance of the diffusion model in offline reinforcement learning tasks.

### 4.1.2 REWARD-BASED DIFFUSION MODULATION

The goal of planning is to generate trajectories that maximize cumulative rewards when executed under the underlying MDP of the given environment. Thus, focusing solely on the fit of the planning trajectories to the transition dynamics is insufficient. It is crucial to guide the diffusion model training to directly align with the planning objective. Therefore, the second auxiliary modulation loss is designed to maximize the reward induced in the generated trajectories. As the trajectories generated from diffusion models do not have reward signals, we predict the reward scores of the state-action pairs $\{(s_t, a_t)\}$ in each trajectory generated through function $\widehat{\boldsymbol{\tau}}_\theta^0(\cdot, \cdot, \cdot)$ using the learned reward function $\widehat{\mathcal{R}}(\cdot, \cdot)$. Specifically, we formulate the reward-based diffusion modulation loss function as the following negative expected trajectory-wise cumulative reward from the generated trajectories:

$$L_{\text{rd}} = -\mathbb{E}_{k \sim \mathcal{U}(1,K), \boldsymbol{\epsilon} \sim \mathcal{N}(\mathbf{0}, \mathbf{I}), \boldsymbol{\tau}^0 \sim \mathcal{D}} \left[ \sum_{(s_t, a_t) \in \widehat{\boldsymbol{\tau}}_\theta^0(\boldsymbol{\tau}^0, k, \boldsymbol{\epsilon})} \widehat{\mathcal{R}}(s_t, a_t) \right] \qquad (10)$$

Through function $\boldsymbol{\tau}_\theta^0$, this loss $L_{\text{rd}}$ again is a function of the diffusion model parameters $\theta$. By computing the expected loss over different time steps $k \in [1 : K]$, different random noise $\boldsymbol{\epsilon}$, and all input trajectories from the offline dataset $\mathcal{D}$, we ensure that the modulation is consistently enforced across all instances of diffusion model training.

By minimizing this reward-based loss, we ensure that the generated trajectories are not only plausible but also reward-optimized to align with the reward structure inherent in the offline data. This approach improves the quality of the trajectories generated from the diffusion model and enhances the overall policy learning process in offline reinforcement learning tasks.

### 4.1.3 REWARD-AWARE DIFFUSION LOSS

In addition to the auxiliary modulation losses, we propose to further align diffusion model training with the goal of RL planning by devising a novel reward-aware diffusion loss to replace the original one. The original diffusion loss (shown in Eq.(5)) minimizes the expected per-trajectory mean squared error between the true additive noise and the predicted noise, which gives equal weights to different training trajectories without differentiation. In contrast, we propose to weight each trajectory instance $\boldsymbol{\tau}^0$ from the offline dataset $\mathcal{D}$ using its normalized cumulative reward, so that the diffusion training can focus more on the more informative trajectory instances with larger cumulative rewards. Specifically, we weight each training trajectory $\boldsymbol{\tau}^0$ using its normalized cumulative reward and formulate the following reward-aware diffusion loss:

$$L_{\text{wdiff}} = \mathbb{E}_{k \sim \mathcal{U}(1,K), \boldsymbol{\epsilon} \sim \mathcal{N}(\mathbf{0}, \mathbf{I}), \boldsymbol{\tau}^0 \sim \mathcal{D}} \left[ \left( \sum_{(s_t, a_t) \in \boldsymbol{\tau}^0} \frac{\mathcal{R}(s_t, a_t)}{T_{\max} \cdot r_{\max}} \right) \left\| \boldsymbol{\epsilon} - \epsilon_\theta(\sqrt{\bar{\alpha}_k} \boldsymbol{\tau}^0 + \sqrt{1 - \bar{\alpha}_k} \boldsymbol{\epsilon}, k) \right\|^2 \right]$$
$$(11)$$

---

**Algorithm 1** Diffusion Training

---

**Require:** Offline data $\mathcal{D} = \{(s_0^i, a_0^i, r_0^i, \ldots, s_T^i, a_T^i, r_T^i)\}$.

Learn transition model $\widehat{\mathcal{T}}(s_t, a_t)$ and reward function $\widehat{\mathcal{R}}(s_t, a_t)$ from offline data $\mathcal{D}$.

Initialize noise network $\epsilon_\theta(\tau^k, k)$.

**while** not converged **do**

    Sample a trajectory from offline data $\boldsymbol{\tau}^0 \sim \mathcal{D}$.

    Sample a random diffusion step $k \sim \mathcal{U}(1, K)$.

    Sample a random noise $\boldsymbol{\epsilon} \sim \mathcal{N}(0, \mathbf{I})$.

    Calculate the gradient $\nabla_\theta L_{\text{total}}$ of Eq. (12) and take gradient descent step.

**end while**

---

Here, $\sum_{(s_t, a_t) \in \boldsymbol{\tau}^0} \mathcal{R}(s_t, a_t)$ is the trajectory-wise cumulative reward on the original offline data instance $\boldsymbol{\tau}^0 \in \mathcal{D}$; $T_{\max}$ denotes the largest trajectory length and $r_{\max}$ denotes the maximum possible per-step reward. By using $T_{\max} \cdot r_{\max}$ as the normalizer, we scale the cumulative reward to a ratio within $(0, 1]$ to weight the corresponding per-trajectory diffusion loss. This weighting mechanism biases the diffusion model toward high-reward trajectories, ensuring that those trajectories yielding higher cumulative rewards are more accurately represented, thus aligning diffusion training with the planning objectives in offline RL. This approach improves the model's performance on rare but valuable trajectories, which are crucial for effective RL.

### 4.1.4 FULL MODULATION FRAMEWORK

The proposed full modulated diffusion model comprises all of the three loss components presented above: the reward-aware diffusion loss $L_{\text{wdiff}}$, the transition-based auxiliary modulation loss $L_{\text{tr}}$, and the reward-based auxiliary modulation loss $L_{\text{rd}}$. By integrating these loss terms together, we have the following total loss for modulated diffusion training:

$$L_{\text{total}} = L_{\text{wdiff}} + \lambda_{\text{tr}} L_{\text{tr}} + \lambda_{\text{rd}} L_{\text{rd}}, \tag{12}$$

where $\lambda_{\text{tr}}$ and $\lambda_{\text{rd}}$ are trade-off parameters that balance the contributions of the transition-based and reward-based auxiliary losses, respectively. Standard diffusion training algorithm can be utilized to train the model $\theta$ by minimizing this total loss function. By employing this integrated loss function, we establish a comprehensive modulation framework that incorporates essential domain and task knowledge into diffusion model training, offering a general capacity of enhancing the adaptation and broadening the applicability of diffusion models.

### 4.1.5 DIFFUSION TRAINING ALGORITHM

The complete training process of the diffusion model is presented in Algorithm 1. Prior to training the diffusion model, a probabilistic transition model $\widehat{\mathcal{T}}(s_t, a_t)$ and a reward model $\widehat{\mathcal{R}}(s_t, a_t)$ are learned from the offline dataset $\mathcal{D}$. Afterward, the noise network is initialized and iteratively trained. During each iteration, an original trajectory $\boldsymbol{\tau}^0$ is sampled from the offline dataset $\mathcal{D}$, along with a randomly selected diffusion step $k$ and noise sample $\boldsymbol{\epsilon}$. Gradient descent is then applied to minimize the total loss $L_{\text{total}}$.

### 4.2 PLANNING WITH DUAL GUIDANCE

Once trained, the diffusion model can be used to generate trajectories for planning during an RL agent's online interactions with the environment. The generation procedure starts from an initial noise trajectory $\boldsymbol{\tau}^K \sim \mathcal{N}(\mathbf{0}, \mathbf{I})$, and gradually denoises it by following the reverse diffusion process $\boldsymbol{\tau}^{k-1} \sim \mathcal{N}(\boldsymbol{\mu}^{k-1}, \sigma_k^2 \mathbf{I})$ for each time step $k \in \{K, K-1, \ldots, 1\}$, where $\boldsymbol{\mu}^{k-1}$ is estimated through Eq. (4). In each diffusion time step $k$, the first state $s_0$ of the trajectory $\boldsymbol{\tau}^k$ is fixed to the current state $s$ of the RL agent in the online environment to ensure the plan starts from it. The denoised trajectory $\boldsymbol{\tau}^0$ after $K$ diffusion time steps is treated as the plan for the RL agent, which is intended to maximize the RL agent's long-term performance without extra interaction with the environment.

To further enhance the objective of planning, some previous work (Janner et al., 2022) has utilized the learned reward function to guide the sampling process of planning. In this work, we propose to

---

**Algorithm 2** Planning with Dual Guidance

---

**Require:** Noise network $\epsilon_\theta$, tradeoff parameter $\alpha$, environment ENV, covariances $\{\sigma_k^2\}$.
  Initialize environment step $t = 0$.
  **while** not finished **do**
    Initialize noise trajectory $\boldsymbol{\tau}_t^K$: $\boldsymbol{\tau}_t^K \sim \mathcal{N}(0, \mathbf{I})$.
    **for** diffusion step $k = K, \ldots, 1$ **do**
      Compute the mean $\mu^{k-1}$ using Eq. (4).
      Compute the guidance $\mathbf{g}$ using Eq. (14).
      Sample next trajectory $\boldsymbol{\tau}_t^{k-1}$ with Eq.(13)
      Set current state $s_t$ to the trajectory: $\boldsymbol{\tau}_t^{k-1}(s_0) = s_t$.
    **end for**
    Execute the first action of plan $\boldsymbol{\tau}_t^0(a_0)$: $s_{t+1} = \text{ENV}(s_t, \boldsymbol{\tau}_t^0(a_0))$
    Increment environment step by 1: $t = t + 1$
  **end while**

---

deploy dual guidance for each reverse diffusion step $k$ by exploiting both the reward function $\widehat{\mathcal{R}}$ and the transition model $\widehat{\mathcal{T}}$ learned from the offline dataset $\mathcal{D}$. Following previous works on conditional reverse diffusion (Dhariwal and Nichol, 2021), we incorporate the dual guidance by perturbing the mean of the Gaussian distribution $\mathcal{N}(\boldsymbol{\mu}^{k-1}, \sigma_k^2 \mathbf{I})$ used for reverse diffusion sampling. Specifically, we integrate the gradient $\mathbf{g}$ of the linear combination of the reward function and transition function w.r.t the trajectory into $\boldsymbol{\mu}^{k-1}$, such that:

$$\boldsymbol{\tau}^{k-1} \sim \mathcal{N}(\boldsymbol{\mu}^{k-1} + \alpha \sigma_k^2 \mathbf{I} \mathbf{g}, \sigma_k^2 \mathbf{I}) \tag{13}$$

where $\mathbf{g}$ is computed as:

$$\mathbf{g} = \sum_{t=0}^{T} \nabla_{(s_t, a_t)} \widehat{\mathcal{R}}(s_t, a_t) + \lambda \sum_{t=0}^{T-1} \nabla_{(s_t, a_t)} \log \widehat{\mathcal{T}}(s_{t+1} | s_t, a_t) \tag{14}$$

where $\alpha$ is a tradeoff parameter that controls the degree of guidance. By incorporating both the reward and transition guidance, we aim to enhance the planning process to generate high-quality trajectories that are both reward-optimized and transition-consistent, improving the overall planning performance. The details of the proposed planning procedure is summarized in Algorithm 2.

## 5 EXPERIMENT

In this section, we present the experimental setup and results for evaluating our proposed method, DMEMM, across various offline RL tasks. We conduct experiments on the D4RL locomotion suite and Maze2D environments to assess the performance of DMEMM compared to several state-of-the-art methods. The experiments are designed to demonstrate the effectiveness of our approach across different tasks, expert levels, and complex navigation scenarios.

**Environments** We conduct our experiments on D4RL (Fu et al., 2020) tasks to evaluate the performance of planning in offline RL settings. Initially, we focus on the D4RL locomotion suite to assess the general performance of our planning methods across different tasks and expert levels of demonstrations. The RL agents are tested on three different tasks: HalfCheetah, Hopper, and Walker2d, and three different levels of expert demonstrations: Med-Expert, Medium, and Med-Replay. We use the normalized scores provided in the D4RL (Fu et al., 2020) benchmarks to evaluate performance. Subsequently, we conduct experiments on Maze2D (Fu et al., 2020) environments to evaluate performance on maze navigation tasks.

**Comparison Methods** We benchmark our methods against several leading approaches in each task domain, including Model Predictive Path Integral (MPPI) (Williams et al., 2016), Batch-Constrained Deep Q-Learning (BCQ) (Fujimoto et al., 2019), Conservative Q-Learning (CQL) (Kumar et al., 2020), Implicit Q-Learning (IQL) (Kostrikov et al., 2022), and Decision Transformer (DT) (Chen et al., 2021).Additionally, we compare our methods with the state-of-the-art offline RL approach, Selecting from Behavior Candidates (SfBC) (Chen et al., 2023), as well as several

Table 1: This table presents the scores on D4RL locomotion suites for various comparison methods. Results are averaged over 5 seeds.

| Gym Tasks | BC | DT | IQL | CQL | SfBC | LDCQ | Diffuser | DD | HDMI | HD-DA | DMEMM (Ours) |
|---|---|---|---|---|---|---|---|---|---|---|---|
| HalfCheetah (Med-Expert) | 55.2 | 86.8 | 86.7 | 91.6 | 92.6±0.5 | 90.2 ± 0.9 | 88.9±0.3 | 90.6±1.3 | 92.1±1.4 | 92.5±0.3 | **94.6±1.2** |
| Hopper (Med-Expert) | 52.5 | 107.6 | 91.5 | 105.4 | 108.6±2.1 | 109.3 ± 0.4 | 103.3±1.3 | 111.8±1.8 | 113.5±0.9 | 115.3±1.1 | **115.9±1.6** |
| Walker2d (Med-Expert) | 107.5 | 108.1 | 109.6 | 108.8 | 109.8±0.2 | 111.3 ± 0.2 | 106.9±0.2 | 108.8±1.7 | 107.9±1.2 | 107.1±0.1 | **111.6±1.1** |
| HalfCheetah (Medium) | 42.6 | 42.6 | 47.4 | 44.0 | 45.9±2.2 | 42.8 ± 0.7 | 42.8±0.3 | 49.1±1.0 | 48.0±0.9 | 46.7±0.2 | **49.2±0.8** |
| Hopper (Medium) | 52.9 | 67.6 | 66.3 | 58.5 | 57.1±4.1 | 69.4 ± 3.5 | 74.3±1.4 | 79.3±3.6 | 76.4±2.6 | 99.3±0.3 | **101.2±1.4** |
| Walker2d (Medium) | 75.3 | 74.0 | 78.3 | 72.5 | 77.9±2.5 | 66.2 ± 1.7 | 79.6±0.6 | 82.5±1.4 | 79.9±1.8 | 84.0±0.6 | **86.5±1.5** |
| HalfCheetah (Med-Replay) | 36.6 | 36.6 | 44.2 | 45.5 | 37.1±1.7 | 41.8 ± 0.4 | 37.7±0.5 | 39.3±4.1 | 44.9±2.0 | 38.1±0.7 | **46.1±1.3** |
| Hopper (Med-Replay) | 18.1 | 82.7 | 94.7 | 95.0 | 86.2±9.1 | 68.5 ± 4.3 | 93.6±0.4 | 100.0±0.7 | 99.6±1.5 | 94.7±0.7 | **100.6±0.9** |
| Walker2d (Med-Replay) | 26.0 | 66.6 | 73.9 | 77.2 | 65.1±5.6 | 86.2 ± 2.5 | 70.6±1.6 | 75.0±4.3 | 80.7±2.1 | 84.1±2.2 | **85.8±2.6** |
| **Average** | 51.9 | 74.7 | 77.0 | 77.6 | 75.6 | 76.2 | 77.5 | 81.8 | 82.6 | 84.6 | **87.9** |

Table 2: This table presents the scores on Maze2D navigation tasks for various comparison methods. Results are averaged over 5 seeds.

| Environment | MPPI | IQL | Diffuser | HDMI | HD-DA | DMEMM (Ours) |
|---|---|---|---|---|---|---|
| Maze2D U-Maze | 33.2 | 47.4 | 113.9±3.1 | 120.1±2.5 | 128.4±3.6 | **132.4±3.0** |
| Maze2D Medium | 10.2 | 34.9 | 121.5±2.7 | 121.8±1.6 | 135.6±3.0 | **138.2±2.2** |
| Maze2D Large | 5.1 | 58.6 | 123.0±6.4 | 128.6±2.9 | **155.8±2.5** | 153.2±3.3 |
| Multi2D U-Maze | 41.2 | 24.8 | 128.9±1.8 | 131.3±1.8 | 144.1±1.2 | **145.6±2.6** |
| Multi2D Medium | 15.4 | 12.1 | 127.2±3.4 | 131.6±1.9 | 140.2±1.6 | **140.8±2.2** |
| Multi2D Large | 8.0 | 13.9 | 132.1±5.8 | 135.4±2.5 | **165.5±0.6** | 159.6±3.8 |
| AntMaze U-Maze | – | 62.2 | 76.0±7.6 | 86.1±2.4 | 94.0±4.9 | **96.2±5.5** |
| AntMaze Medium | – | 70.0 | 31.9±5.1 | – | 88.7±8.1 | **90.1±6.4** |
| AntMaze Large | – | 47.5 | 0.0±0.0 | 71.5±3.5 | **83.6±5.8** | 79.6±7.7 |

diffusion-based offline RL methods, including Diffuser (Janner et al., 2022), Decision Diffuser (DD) (Ajay et al., 2023), Latent Diffusion-Constrained Q-learning (LDCQ / LDGC) (Venkatraman et al., 2024), Hierarchical Diffusion for Offline Decision Making (HDMI) (Li et al., 2023), and Hierarchical Diffuser with Dense Actions (HD-DA) (Chen et al., 2024).

## 5.1 EXPERIMENTAL RESULTS ON D4RL

The experimental results summarized in Table 1 highlight the performance of various comparison methods across different Gym tasks, with scores averaged over 5 seeds. Our proposed method, DMEMM, consistently outperforms other methods across all tasks. Notably, in the HalfCheetah environments, DMEMM achieves a 2.0-point improvement on the Med-Expert dataset, and an 6.8-point improvement on the Med-Replay dataset compared to the previous best results. Additionally, DMEMM shows a 2.8-point increase on the Med-Expert Walker2D task, demonstrating that DMEMM effectively extracts valuable information, particularly from data that is not purely expert-level.

In most tasks, DMEMM outperforms the previous state-of-the-art method HD-DA, another variant of a Diffuser based planning method, by more than 2.0 points on average. Compared to Diffuser, DMEMM shows superior performance on all tasks, indicating that our method improves the consistency and optimality of diffusion model training in offline RL planning.

Overall, DMEMM achieves outstanding performance. With an average score of 87.9, DMEMM yields a substantial improvement over the second-highest average score of 84.6 achieved by HD-DA. These results clearly demonstrate the robustness and superiority of DMEMM in enhancing performance across various Gym tasks.

## 5.2 EXPERIMENTAL RESULTS ON MAZE2D

We present our experimental results on the Maze2D navigation tasks in Table 2, where the results are averaged over 5 seeds. The table shows that in all three environments, particularly at the U-Maze and Medium difficulty levels, our proposed DMEMM method significantly outperforms other comparison methods. Specifically, on Maze2D tasks, DMEMM achieves a 4.0 point improvement over the state-of-the-art HD-DA method on the U-Maze task, and a 2.6 point increase on the Medium-sized maze. Compared to Diffuser, DMEMM shows an almost 20-point improvement. These results indicate that our method performs exceptionally well in generating planning solutions for navigation tasks.

Table 3: the scores on the Walker2D environment at three different levels for all four ablation variants. Results are averaged over 5 seeds.

| Gym Tasks | DMEMM | DMEMM-w/o-weighting | DMEMM-w/o-$\lambda_{tr}$ | DMEMM-w/o-$\lambda_{rd}$ | DMEMM-w/o-tr-guide |
|---|---|---|---|---|---|
| Med-Expert | **111.6±1.1** | 110.4±0.8 | 108.4±1.2 | 110.4±0.6 | 109.9±1.0 |
| WMedium | **86.5±1.5** | 85.6±1.2 | 82.8±1.4 | 84.4±0.9 | 83.0±1.8 |
| Med-Replay | **85.8±2.6** | 84.6±2.2 | 82.2±1.7 | 83.7±2.5 | 82.6±3.2 |

However, HD-DA shows better performance on the large maze tasks. This is likely due to the hierarchical structure of HD-DA, which offers an advantage in larger, more complex environments by breaking long-horizon planning into smaller sub-tasks, an area where our method is not specifically designed to excel. Nevertheless, DMEMM remains competitive in larger environments, while demonstrating superior performance in smaller and medium-sized tasks.

## 5.3 ABLATION STUDY

We conduct an ablation study to evaluate the effectiveness of the key components in our DMEMM framework. We compare the full DMEMM model with four ablated variants: (1) DMEMM-w/o-weighting, which removes the weighting function in the reward-aware diffusion loss; (2) DMEMM-w/o-$\lambda_{\text{tr}}$, which omits the transition-based diffusion modulation loss; (3) DMEMM-w/o-$\lambda_{\text{rd}}$, which omits the reward-based diffusion modulation loss; and (4) DMEMM-w/o-tr-guide, which removes transition guidance in the dual-guided sampling procedure. The ablation study is conducted on all locomotion tasks across three levels of expert demonstrations, while only the Walker2D results are reported in the main paper. The complete results are provided in Table 5 of the Appendix. Table 3 summarizes the performance of all four ablation variants on the D4RL locomotion benchmarks, averaged over five random seeds.

The results highlight the contribution of each component in DMEMM. Across all three difficulty levels, the full DMEMM model consistently achieves the best performance. In particular, removing transition-related components, either the transition-based modulation loss (DMEMM-w/o-$\lambda_{\text{tr}}$) or the transition guidance (DMEMM-w/o-tr-guide), leads to substantial performance drops, underscoring the importance of explicitly modeling transition dynamics in our approach. Incorporating transition information significantly improves the consistency and fidelity of generated trajectory plans. Moreover, DMEMM-w/o-$\lambda_{\text{rd}}$ and DMEMM-w/o-weighting yield comparable results, with DMEMM-w/o-$\lambda_{\text{rd}}$ showing a slightly larger degradation. This indicates that the designed reward model and its weighting mechanism play a key role in improving the optimality of planned trajectories.

Overall, the ablation study demonstrates that each component of our DMEMM method contributes significantly to its performance. Removing any of these components results in a noticeable decrease in performance, highlighting the importance of the weighting function, transition-based and reward-based diffusion modulation loss, and transition guidance in achieving optimal results in offline reinforcement learning tasks.

## 6 CONCLUSION

In this work, we addressed a critical limitation of conventional diffusion-based planning methods in offline RL, which often overlook the consistency of transition dynamics in planned trajectories. To overcome this challenge, we proposed Diffusion Modulation via Environment Mechanism Modeling (DMEMM), a novel approach that integrates RL-specific environment mechanisms, particularly transition dynamics and reward functions, into the diffusion model training process. By modulating the diffusion loss with cumulative rewards and introducing auxiliary losses based on transition dynamics and reward functions, DMEMM enhances both the coherence and quality of the generated trajectories, ensuring they are plausible and optimized for policy learning. Our experimental results across multiple offline RL environments demonstrate the effectiveness of DMEMM, achieving state-of-the-art performance compared to previous diffusion-based planning methods. The proposed approach significantly improves the alignment of generated trajectories, addressing the discrepancies between offline data and real-world environments. This provides a promising framework for further exploration of diffusion models in RL and their potential practical applications.

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

# A PROOF OF PROPOSITION 1

In this section, we present the proof of Proposition 1.

*Proof.* To incorporate key RL mechanisms into the training of the diffusion model, we explore the denoising process and trace the denoised data through the reverse diffusion process. Let $\hat{\tau}^0$ represent the denoised output trajectory. It can be gradually denoised using the reverse process, following the chain rule: $\hat{\tau}^0 \sim p_\theta(\tau^K) \prod_{k=1}^K p_\theta(\tau^{k-1}|\tau^k)$, where the detailed reverse process is defined in Eq. (3) and Eq. (4). Starting from an intermediate trajectory $\tau^k$ at step $k$, by combining these two equations, the trajectory at the next diffusion step, $k-1$, can be directly sampled from the distribution:

$$\hat{\tau}^{k-1} \sim \mathcal{N}\left(\frac{1}{\sqrt{\alpha_k}}\left(\tau^k - \frac{1-\alpha_k}{\sqrt{1-\bar{\alpha}_k}}\epsilon_\theta(\tau^k, k)\right), \sigma_k^2 \mathbf{I}\right). \tag{15}$$

By applying the reparameterization trick (Kingma and Welling, 2014), we can derive a closed-form solution for the above distribution. Let $\epsilon_k$ represent the noise introduced in the reverse process $p_\theta(\tau^{k-1}|\tau_k)$, and the denoised trajectory can then be formulated as:

$$\begin{aligned}
\hat{\tau}^{k-1} &= \frac{1}{\sqrt{\alpha_k}}\left(\tau^k - \frac{1-\alpha_k}{\sqrt{1-\bar{\alpha}_k}}\epsilon_\theta(\tau^k, k)\right) + \sigma_k\epsilon_k \\
&= \frac{1}{\sqrt{\alpha_k}}\tau^k - \frac{1-\alpha_k}{\sqrt{(1-\bar{\alpha}_k)\alpha_k}}\epsilon_\theta(\tau^k, k) + \sigma_k\epsilon_k.
\end{aligned} \tag{16}$$

In the following diffusion step $k-2$, the denoised data $\hat{\tau}^{k-2}$ is sampled from a similar Gaussian distribution. By the Central Limit Theorem, $\hat{\tau}^{k-1}$ serves as an unbiased estimate of $\tau^{k-1}$. Therefore, the denoised data $\hat{\tau}^{k-2}$ can be expressed as follows:

$$\begin{aligned}
\hat{\tau}^{k-2} &\sim \mathcal{N}\left(\frac{1}{\sqrt{\alpha_{k-1}}}\left(\tau^{k-1} - \frac{1-\alpha_{k-1}}{\sqrt{1-\bar{\alpha}_{k-1}}}\epsilon_\theta(\tau^{k-1}, k-1)\right), \sigma_{k-1}^2\mathbf{I}\right) \\
&= \frac{1}{\sqrt{\alpha_{k-1}}}\left(\hat{\tau}^{k-1} - \frac{1-\alpha_{k-1}}{\sqrt{1-\bar{\alpha}_{k-1}}}\epsilon_\theta(\tau^{k-1}, k-1)\right) + \sigma_{k-1}\epsilon_{k-1} \\
&= \frac{1}{\sqrt{\alpha_{k-1}}}\left(\frac{1}{\sqrt{\alpha_k}}\tau^k - \frac{1-\alpha_k}{\sqrt{(1-\bar{\alpha}_k)\alpha_k}}\epsilon_\theta(\tau^k, k) - \frac{1-\alpha_{k-1}}{\sqrt{1-\bar{\alpha}_{k-1}}}\epsilon_\theta(\tau^{k-1}, k-1) + \sigma_k\epsilon_k\right) \\
&\quad + \sigma_{k-1}\epsilon_{k-1} \\
&= \frac{1}{\sqrt{\alpha_k\alpha_{k-1}}}\tau^k - \frac{1-\alpha_k}{\sqrt{(1-\bar{\alpha}_k)\alpha_k\alpha_{k-1}}}\epsilon_\theta(\tau^k, k) - \frac{1-\alpha_{k-1}}{\sqrt{(1-\bar{\alpha}_{k-1})\alpha_{k-1}}}\epsilon_\theta(\tau^{k-1}, k-1) \\
&\quad + \frac{1}{\sqrt{\alpha_{k-1}}}\sigma_k\epsilon_k + \sigma_{k-1}\epsilon_{k-1}.
\end{aligned} \tag{17}$$

The introduced noise $\epsilon_{k-1}$ in diffusion step $k-1$ can be combined with the noise $\epsilon_k$ at diffusion step $k$ into a joint noise term, $\bar{\epsilon}_{k-1}$, by merging two Gaussian distributions, $\mathcal{N}(0, \frac{\sigma_k^2}{\alpha_{k-1}}\mathbf{I})$ and $\mathcal{N}(0, \sigma_{k-1}^2\mathbf{I})$, into $\mathcal{N}(0, (\frac{\sigma_k^2}{\alpha_{k-1}} + \sigma_{k-1}^2)\mathbf{I})$. Consequently, we obtain the distribution for the denoised data $\hat{\tau}^{k-2}$ with only directly computable terms, where

$$\begin{aligned}
\hat{\tau}^{k-2} &= \frac{1}{\sqrt{\alpha_k\alpha_{k-1}}}\tau^k - \frac{1-\alpha_k}{\sqrt{(1-\bar{\alpha}_k)\alpha_k\alpha_{k-1}}}\epsilon_\theta(\tau^k, k) - \frac{1-\alpha_{k-1}}{\sqrt{(1-\bar{\alpha}_{k-1})\alpha_{k-1}}}\epsilon_\theta(\tau^{k-1}, k-1) \\
&\quad + \sqrt{\frac{\sigma_k^2}{\alpha_{k-1}} + \sigma_{k-1}^2}\,\bar{\epsilon}_{k-1} \\
&\sim \mathcal{N}\left(\frac{1}{\sqrt{\alpha_k\alpha_{k-1}}}\tau^k - \frac{1-\alpha_k}{\sqrt{(1-\bar{\alpha}_k)\alpha_k\alpha_{k-1}}}\epsilon_\theta(\tau^k, k) - \frac{1-\alpha_{k-1}}{\sqrt{(1-\bar{\alpha}_{k-1})\alpha_{k-1}}}\epsilon_\theta(\tau^{k-1}, k-1),\right. \\
&\quad \left.\left(\frac{\sigma_k^2}{\alpha_{k-1}} + \sigma_{k-1}^2\right)\mathbf{I}\right).
\end{aligned} \tag{18}$$

By repeating the denoising process for $k$ iterations, we can ultimately obtain a closed-form representation of the denoised data $\widehat{\boldsymbol{\tau}}^0$.

$$
\widehat{\boldsymbol{\tau}}^0 = \frac{1}{\sqrt{\prod_{i=1}^{k}\alpha_i}}\boldsymbol{\tau}^k - \sum_{i=1}^{k}\frac{1-\alpha_i}{\sqrt{(1-\bar{\alpha}_i)\prod_{j=1}^{i}\alpha_j}}\epsilon_\theta(\boldsymbol{\tau}^i, i) + \sqrt{\sigma_1^2 + \sum_{i=2}^{k}\frac{\sigma_i^2}{\prod_{j=1}^{i-1}\alpha_j}}\bar{\epsilon}_1
$$

$$
= \frac{1}{\sqrt{\bar{\alpha}_k}}\boldsymbol{\tau}^k - \sum_{i=1}^{k}\frac{1-\alpha_i}{\sqrt{(1-\bar{\alpha}_i)\bar{\alpha}_i}}\epsilon_\theta(\boldsymbol{\tau}^i, i) + \sqrt{\sigma_1^2 + \sum_{i=2}^{k}\frac{\sigma_i^2}{\bar{\alpha}_{i-1}}}\bar{\epsilon}_1. \tag{19}
$$

Using the closed-form representation of the reparameterization trick, the final denoised data $\widehat{\boldsymbol{\tau}}^0$ follows a Gaussian distribution, expressed as $\widehat{\boldsymbol{\tau}}^0 \sim \mathcal{N}(\widehat{\mu}_\theta(\boldsymbol{\tau}^k, k), \widehat{\sigma}^2\mathbf{I})$. The mean $\widehat{\mu}_\theta(\boldsymbol{\tau}^k, k)$ captures the denoising trajectory and is formulated as:

$$
\widehat{\mu}_\theta(\boldsymbol{\tau}^k, k) = \frac{1}{\sqrt{\bar{\alpha}_k}}\boldsymbol{\tau}^k - \sum_{i=1}^{k}\frac{1-\alpha_i}{\sqrt{(1-\bar{\alpha}_i)\bar{\alpha}_i}}\epsilon_\theta(\boldsymbol{\tau}^i, i). \tag{20}
$$

Similarly, the covariance $\widehat{\sigma}^2$ accounts for the accumulation of noise over all diffusion steps and is written as:

$$
\widehat{\sigma}^2 = \sigma_1^2 + \sum_{i=2}^{k}\frac{\sigma_i^2}{\bar{\alpha}_{i-1}}. \tag{21}
$$

$\square$

# B  IMPLEMENTATION DETAILS

**Reward model and transition model pretraining**  Before training the diffusion model, we first pretrain both the reward model and the transition model on the concatenated inputs $(s_t, a_t)$ from the same dataset used for diffusion training (e.g., D4RL (Fu et al., 2020)), with the same Gaussian normalization. The transition model is implemented as a MLP with two hidden layers of 512 units, ReLU activations, and a linear output head predicting the next-state mean $\mu(s_t, a_t) \in \mathbb{R}^{s_{\dim}}$, trained with mean squared error against $s_{t+1}$. The reward model is also an MLP with two hidden layers of 256 units, ReLU activations, and a linear output head, trained by regression to the per-timestep rewards in the dataset. Both models are pretrained using the Adam optimizer (learning rate $3 \times 10^{-4}$), batch size 64, for $5 \times 10^5$ training steps. After pretraining, the reward and transition models are frozen during diffusion model training.

**Diffusion training**  We adopt the core diffusion model and reward guidance implementation from Diffuser (Janner et al., 2022). Both the diffusion backbone and the reward-guidance network use a temporal U-Net trained on length-$T$ trajectories of concatenated $(s_t, a_t)$, with hard conditioning on the initial observation $s_0$. We set the planning horizon to $T = 32$ for locomotion tasks, $T = 128$ for the three U-Maze tasks, $T = 256$ for the three Medium-Maze tasks, and $T = 384$ for the three Large-Maze tasks. Observations and actions are Gaussian-normalized using statistics from the offline dataset.

During training, the diffusion model is optimized with our designed total loss $L_{\text{total}} = L_{\text{wdiff}} + \lambda_{\text{tr}}L_{\text{tr}} + \lambda_{\text{rd}}L_{\text{rd}}$, as defined in Eq. (12). The weights in the reward-aware diffusion loss $L_{\text{wdiff}}$ are clipped by $r_{\max}$, which we set to 1 in practice. We use $\lambda_{\text{tr}} = 0.1$ for the transition-based auxiliary modulation loss $L_{\text{tr}}$ and $\lambda_{\text{rd}} = 0.05$ for the reward-based auxiliary modulation loss $L_{\text{rd}}$. When a domain lacks stepwise rewards, the reward bias term is omitted. The diffusion backbone is trained with the Adam optimizer (learning rate $2 \times 10^{-4}$), batch size 32, gradient accumulation factor 2, and EMA decay 0.995.

**Diffusion sampling**  Following Janner et al. (2022), we use $N = 20$ reverse diffusion steps and apply reward-only gradient guidance with scale $\alpha = 10^{-3}$ at each step, re-imposing conditioning after every step. We report the top-scoring trajectories under reward guidance.

**Computational resources**  All experiments were conducted on a cluster of 10 nodes, each equipped with four Intel Xeon CPUs, 32 GB of RAM, and an NVIDIA GeForce RTX 2080 GPU with 11 GB of VRAM.

Table 4: Quarter-wise transition mismatch $E_i$ (mean squared state prediction error) for Diffuser and DMEMM on D4RL locomotion tasks (Medium-Expert). Results are averaged over 5 runs.

| Method | Environment | Quarter 1 | Quarter 2 | Quarter 3 | Quarter 4 |
|--------|-------------|-----------|-----------|-----------|-----------|
| Diffuser | HalfCheetah | $29.10 \pm 0.32$ | $34.67 \pm 0.40$ | $34.96 \pm 0.37$ | $38.71 \pm 0.42$ |
| DMEMM | HalfCheetah | $\mathbf{9.74 \pm 0.05}$ | $\mathbf{9.88 \pm 0.04}$ | $\mathbf{9.91 \pm 0.09}$ | $\mathbf{9.99 \pm 0.06}$ |
| Diffuser | Hopper | $1.01 \pm 0.06$ | $0.96 \pm 0.05$ | $1.35 \pm 0.08$ | $2.96 \pm 0.12$ |
| DMEMM | Hopper | $\mathbf{0.82 \pm 0.07}$ | $\mathbf{0.75 \pm 0.06}$ | $\mathbf{1.02 \pm 0.09}$ | $\mathbf{1.85 \pm 0.15}$ |
| Diffuser | Walker2D | $1.42 \pm 0.07$ | $3.21 \pm 0.06$ | $3.52 \pm 0.11$ | $4.03 \pm 0.13$ |
| DMEMM | Walker2D | $\mathbf{1.12 \pm 0.08}$ | $\mathbf{2.19 \pm 0.10}$ | $\mathbf{2.43 \pm 0.11}$ | $\mathbf{2.61 \pm 0.14}$ |

## C EVALUATION OF TRANSITION MISMATCH

To investigate the potential transition mismatch problem in conventional Diffuser models (Janner et al., 2022), we conduct an experiment to quantify the discrepancy between trajectories predicted by the diffusion model and the actual environment rollouts. This analysis highlights how traditional Diffuser suffers from model–environment dynamics gaps, and how our proposed DMEMM effectively tackles this issue. We compare Diffuser and DMEMM on D4RL locomotion tasks (Fu et al., 2020) at the Medium-Expert level.

We use the trained diffusion models with reward guidance from both Diffuser and DMEMM for evaluation. During the online sampling stage, the planner at each timestep $t$ generates an imagined future sequence $\{\hat{s}_{t+1}^t, \hat{s}_{t+2}^t, \ldots, \hat{s}_{t+H}^t\}$, where $H$ is the planning horizon. Each predicted state $\hat{s}_{t+h}^t$ is compared to the corresponding ground-truth state $s_{t+h}$ collected from the environment for $h \in \{1, \ldots, H\}$. This measures the discrepancy caused by the mismatch between the diffusion model and the true environment dynamics. At each prediction step we compute the L2-norm error $e_{t,h} = \left\| \hat{s}_{t+h}^t - s_{t+h} \right\|_2^2$. To reduce computation, we reuse previously generated plans by backtracking from stored plans at earlier states rather than recomputing forward rollouts from scratch.

To better analyze prediction quality over different time scales, we divide the planning horizon into four equal-length quarters and report the average error in each quarter. Early quarters reflect short-term prediction accuracy, while later quarters capture long-horizon stability. Formally, let the trajectory length be $T$, the horizon $H$, and let quarter $i \in \{1, 2, 3, 4\}$ cover prediction steps $h \in \left[ \frac{(i-1)H}{4} + 1, \frac{iH}{4} \right]$. The average squared error for $i$-th quarter is:

$$E_i = \frac{1}{(T - H) \cdot \frac{H}{4}} \sum_{t=0}^{T-H-1} \sum_{h=(i-1)H/4+1}^{iH/4} \left\| \hat{s}_{t+h}^t - s_{t+h} \right\|_2^2. \tag{22}$$

These quarter-wise errors quantify how transition mismatch accumulates along the trajectory: smaller $E_i$ in later quarters indicates better long-horizon predictive ability.

The results are summarized in Table 4. We observe that DMEMM greatly outperforms Diffuser in terms of transition mismatch on the HalfCheetah environment. On the other two environments, DMEMM still surpasses Diffuser, though with a smaller relative improvement. We hypothesize that transition mismatch is more severe in complex dynamical systems such as HalfCheetah, suggesting that our method is particularly beneficial for environments with more challenging dynamics.

Furthermore, as the planning horizon increases, Diffuser's prediction errors grow substantially, indicating poor generalization to real online interactions despite strong offline fitting. In contrast, DMEMM consistently maintains lower transition mismatch, especially in the later quarters, without exhibiting the pronounced error escalation seen in Diffuser. Interestingly, although HalfCheetah shows the highest absolute prediction errors, neither Diffuser nor DMEMM displays a sharp error increase across quarters in this task.

Overall, these findings demonstrate the superior long-horizon prediction quality and robustness of the proposed DMEMM method.

Table 5: This table presents the scores on D4RL locomotion suites for all four ablation variants. Results are averaged over 5 seeds.

| Gym Tasks | DMEMM | DMEMM-w/o-weighting | DMEMM-w/o-$\lambda_{tr}$ | DMEMM-w/o-$\lambda_{rd}$ | DMEMM-w/o-tr-guide |
|---|---|---|---|---|---|
| HalfCheetah (Med-Expert) | **94.6±1.2** | 93.8±0.9 | 92.2±0.6 | 92.8±1.2 | 92.5±1.3 |
| Hopper (Med-Expert) | **115.9±1.6** | 115.2±0.4 | 114.4±0.8 | 115.0±0.4 | 114.8±0.2 |
| Walker2d (Med-Expert) | **111.6±1.1** | 110.4±0.8 | 108.4±1.2 | 110.4±0.6 | 109.9±1.0 |
| HalfCheetah (Medium) | **49.2±0.8** | 48.0±1.1 | 46.3±0.4 | 47.1±0.6 | 46.9±0.9 |
| Hopper (Medium) | **101.2±1.4** | 100.4±1.2 | 98.6±1.8 | 100.1±1.1 | 99.8±1.6 |
| Walker2d (Medium) | **86.5±1.5** | 85.6±1.2 | 82.8±1.4 | 84.4±0.9 | 83.0±1.8 |
| HalfCheetah (Med-Replay) | **46.1±1.3** | 44.7±1.7 | 42.5±2.9 | 44.2±1.4 | 43.6±2.5 |
| Hopper (Med-Replay) | **100.6±0.9** | 98.8±1.2 | 97.0±0.9 | 98.2±0.6 | 96.2±1.2 |
| Walker2d (Med-Replay) | **85.8±2.6** | 84.6±2.2 | 82.2±1.7 | 83.7±2.5 | 82.6±3.2 |

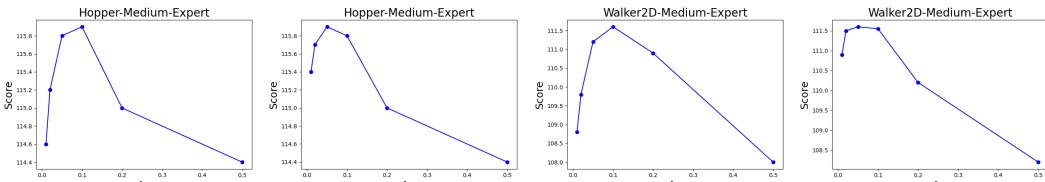

Figure 1: Hyperparameter sensitivity analysis of the tradeoff parameters for transition-based diffusion modulation loss ($\lambda_{tr}$) and reward-based diffusion modulation loss ($\lambda_{rd}$) on Hopper-Medium-Expert and Walker2D-Medium-Expert environments.

## D    FULL ABLATION RESULTS

The complete ablation results of our DMEMM method compared with the four ablated variants (1) DMEMM-w/o-weighting, which removes the weighting function in the reward-aware diffusion loss; (2) DMEMM-w/o-$\lambda_{\text{tr}}$, which omits the transition-based diffusion modulation loss; (3) DMEMM-w/o-$\lambda_{\text{rd}}$, which omits the reward-based diffusion modulation loss; and (4) DMEMM-w/o-tr-guide, which removes transition guidance in the dual-guided sampling procedure are presented in Table 5.

The conclusions drawn from the full ablation results are consistent with those reported in the main paper. Across all three environments and all expert demonstration levels, the performance of DMEMM is substantially degraded when either the transition-based diffusion modulation loss or the transition guidance is removed, highlighting the critical role of explicitly modeling transition dynamics. Dropping the weighting function or the reward-based diffusion modulation loss also harms performance, with the reward auxiliary loss $L_{\text{rd}}$ appearing relatively more important between the two. Overall, removing any single component leads to a noticeable performance drop compared to the full DMEMM model, demonstrating the effectiveness and necessity of each key component in our approach.

## E    HYPERPARAMETER SENSITIVITY ANALYSIS

In this section, we analyze the sensitivity of the tradeoff parameters $\lambda_{\text{tr}}$ (transition-based diffusion modulation loss) and $\lambda_{\text{rd}}$ (reward-based diffusion modulation loss) to understand their impact on performance in offline RL tasks. The analysis is conducted on two environments: Hopper-Medium-Expert and Walker2D-Medium-Expert.

Figures 1 illustrate the performance sensitivity to the tradeoff parameters. For $\lambda_{\text{tr}}$, the performance peaks at approximately $\lambda_{\text{tr}} = 0.1$ in both the Walker2D-Medium-Expert and Hopper-Medium-Expert environments. Beyond this optimal point, performance declines notably, regardless of whether $\lambda_{\text{tr}}$ is increased or decreased. Similarly, for $\lambda_{\text{rd}}$, the performance also peaks around $\lambda_{\text{rd}} = 0.05$ in both environments. However, unlike $\lambda_{\text{tr}}$, performance shows little change when $\lambda_{\text{rd}}$ is adjusted within a small range, indicating that $\lambda_{\text{rd}}$ is less sensitive than $\lambda_{\text{tr}}$. Overall, the hyperparameter sensitivity analysis shows that both $\lambda_{\text{rd}}$ and $\lambda_{\text{tr}}$ have similar effects on performance and are robust across different tasks. Additionally, it confirms that the selected hyperparameters for our experiments are optimal.

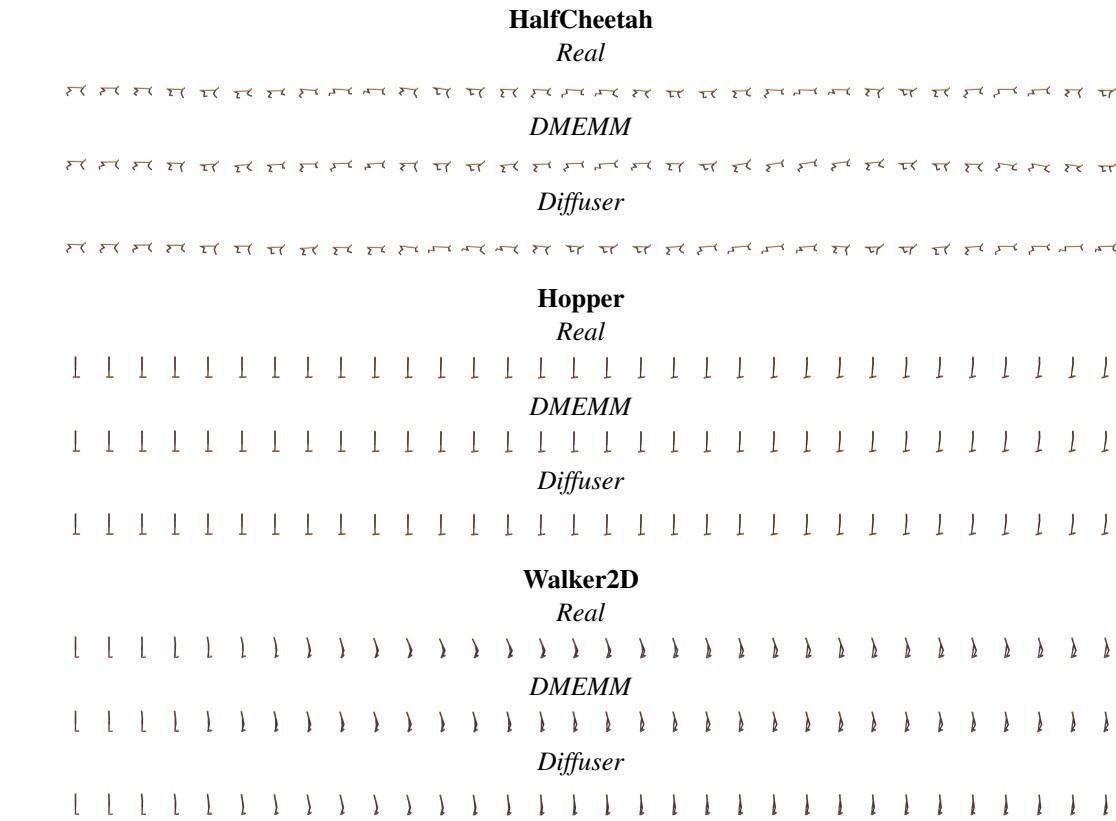

Figure 2: Visualization of trajectories generated by Real environment rollouts, DMEMM, and Diffuser on HalfCheetah, Hopper, and Walker2D (32-long horizon per method). Figures are slightly overlapped horizontally to highlight differences in posture and stability.

## F    TRAJECTORY CONSISTENCY VISUALIZATION

We provide a qualitative visualization of the trajectories generated during online planning to demonstrate that our proposed DMEMM method improves transition consistency. Starting from a fixed initial state, we sample trajectories using DMEMM and the standard Diffuser, and compare them against ground-truth trajectories collected from the offline dataset. To ensure a fair comparison, we visualize 32 consecutive frames for each method. The results are shown in Figure E.

From the figure, we observe that DMEMM produces trajectories that closely resemble the real trajectories across all environments, especially on HalfCheetah. For Hopper, the trajectories generated by all methods are visually similar, which is consistent with the quantitative results in Table 4. On Walker2D, trajectories generated by DMEMM remain closer to the real trajectories than those produced by Diffuser, indicating that DMEMM can generate more consistent and stable motions than the base Diffuser method.

