# OpenReview forum: "Diffusion Modulation via Environment Mechanism Modeling for Planning"
_ICLR.cc/2026/Conference — Submitted to ICLR 2026_

### Official Review · Reviewer_vn9X · 2025-10-19

**Soundness:** 3
**Presentation:** 2
**Contribution:** 2
**Rating:** 4
**Confidence:** 4

**Summary:**

The author proposes DMEMM, a approach for trajectory generation in offline reinforcement learning which explicitly enforce consistency with the underlying environment dynamics. DMEMM integrates transition dynamics and reward functions into diffusion model training through three main components (a reward-aware diffusion loss, a transition-based auxiliary loss and a reward-based auxiliary loss). Experiments on D4RL locomotion and Maze2D tasks demonstrate effectiveness of DMEMM.

**Strengths:**

1. Experimental results demonstrate that DMEMM performs well on D4RL tasks.
2. The motivation is highly intuitive and reasonable.

**Weaknesses:**

1. The experiments rely solely on the D4RL benchmark, which has become saturated for many locomotion and maze-solving tasks. Evaluating the method on more recent or challenging benchmarks (e.g., OGBench) would better demonstrate its scalability and robustness.
2. The strong results are demonstrated on tasks with relatively low-dimensional state and action spaces. A key component of DMEMM is the explicit transition model which is known to be challenging to scale to high-dimensional state or action spaces (e.g., OGBench humanoid maze). This raises a critical question about the method's applicability to these more complex domains.
3. The evaluation is purely quantitative. The paper would be significantly strengthened by including visualizations that compare trajectories generated by DMEMM against baselines.
4. Although this paper focuses on training-time dynamic consistency improvement, I suggest the authors include more related works on inference-time improvement methods. Here are some relevant papers:

[1] SafeDiffuser: Safe Planning with Diffusion Probabilistic Models

[2] Refining Diffusion Planner for Reliable Behavior Synthesis by Automatic Detection of Infeasible Plans

[3] Resisting stochastic risks in diffusion planners with the trajectory aggregation tree

[4] Inference-Time Policy Steering through Human Interactions

[5] Local Manifold Approximation and Projection for Manifold-Aware Diffusion Planning

**Questions:**

see weaknesses

---

> ### Author Response · Authors · 2025-11-21
>
> We thank the reviewer for the detailed comments and thoughtful suggestions.
>
> **Q1.** About the benchmark.
>
> **Answer:** We selected D4RL because it is a widely accepted benchmark for offline RL, especially for diffusion-based RL methods. Most prior works evaluate on D4RL, which allows fair comparison across methods. We considered OGBench; however, OGBench is specifically designed for goal-conditioned offline RL, whereas our work and the baseline offline RL methods are reward-based and require reward signals. In addition, many comparison methods do not report results on OGBench, making it difficult for us to conduct a fair and complete evaluation.
>
> **Q2.** About the transition model.
>
> **Answer:** Although some locomotion tasks in D4RL are low-dimensional, we have also evaluated our method on higher-dimensional environments. For example, AntMaze has a 111-dimensional state input, which is even higher than some “high-dimensional” tasks such as OGBench Humanoid. Our method performs well on AntMaze, demonstrating its extensibility to more complex tasks. Also, we use the transition model only for one-step prediction rather than long-horizon rollout, which is less affected by high-dimensional inputs. With the help of our transition loss, DMEMM further improves its handling of high-dimensional dynamics.
>
> **Q3.** About visualization of trajectories.
>
> **Answer:** We reported quantitative improvements in transition precision in Table 4 of the Appendix. We thank the reviewer for the suggestion of visualizing trajectories generated by DMEMM vs. the baseline. We will include trajectory visualizations in the revised version.
>
> **Q4.** About citations for inference-time methods.
>
> **Answer:** As our method mainly focuses on modifying the training procedure rather than inference-time acceleration, we did not include inference-time methods. We thank the reviewer for the suggestions on related work, and we will carefully include the most relevant citations in the revision.

---

> > ### Comment · Reviewer_vn9X · 2025-11-24
> >
> > Thank you for the response from the authors.
> >
> > **Regarding Q1**, while I agree that OGBench is designed for goal-conditioned RL, I do not fully agree with the statement that “many comparison methods do not report results on OGBench,” as several recent diffusion-based rl works such as [1–4] do provide OGBench results. Given the relatively long rebuttal period, could the authors additionally report results on more challenging and realistic domains such as manipulation tasks?
> >
> > **Regarding Q2**, my understanding is that AntMaze generally has a state dimension of around 29 with an 8-dimensional action space, which is not particularly high-dimensional. Thus, claim about scalability to high-dimensional or more complex domains does not seem sufficiently supported.
> >
> > **Regarding Q3**, it is indeed possible to include trajectory visualizations in the paper. Simply stating that visualizations will be added in the revised version does not fully resolve the concern about qualitative validation.
> >
> > Overall, several concerns remain insufficiently addressed, so I will maintain my score.
> >
> > [1] Monte Carlo Tree Diffusion for System 2 Planning
> >
> > [2] Generative Trajectory Stitching through Diffusion Composition
> >
> > [3] State-Covering Trajectory Stitching for Diffusion Planners
> >
> > [4] Elastic-Subgoal Diffused Policy Learning

---

> > > ### Author Response · Authors · 2025-12-04
> > >
> > > We thank the reviewer for the follow-up discussion and provide responses to address the additional concerns.
> > >
> > > **Q5.** About experiments on more complex tasks.
> > >
> > > **Answer:**
> > > First, we would like to reiterate that OGBench may not be a perfect fit for our evaluation setting. The core environments in OGBench are primarily designed for goal-conditioned offline RL, where rewards are sparse and require specialized mechanisms to solve effectively. The methods referenced by the reviewer, including two trajectory stitching approaches [1, 2] and the replanning-based MCTD method [3], are specifically tailored for long-horizon goal-conditioned planning and demonstrate strong performance in that setting. In fact, the trajectory stitching works evaluate their methods only against goal-conditioned offline RL baselines, making a direct comparison on OGBench less meaningful for our setting.
> > >
> > > While our method is able to perform long-horizon planning, it is primarily designed for **traditional reward-dense tasks**, rather than goal-conditioned sparse-reward problems.
> > >
> > > Second, we nonetheless conducted a preliminary evaluation on two OGBench manipulation tasks, cube-single-play-v0 and cube-double-play-v0, which involve robotic cube rearrangement. We chose these environments because they are more complex and not covered in our earlier experiments. Results are summarized in the table below. Each entry reports the success rate of completing the task. Results for Diffusion Forcing and MCTD are taken from [3]. For Diffuser and DMEMM, we perform per-step planning by executing the first action from each planning horizon.
> > >
> > > | Dataset          | Diffuser           | Diffusion Forcing     | MCTD               | DMEMM (Ours)               |
> > > |------------------|--------------------|------------------------|---------------------------|--------------------|
> > > | cube-single-play-v0   | $95 \pm 12$      | $\mathbf{100 \pm 0}$ | $\mathbf{100 \pm 0}$    | $\mathbf{100 \pm 0}$ |
> > > | cube-double-play-v0   | $13 \pm 10$      | $18 \pm 11$          | $\mathbf{50 \pm 16}$             | $16 \pm 11$ |
> > >
> > > As shown, DMEMM achieves perfect performance on cube-single-play, matching the best-performing methods. On the more challenging cube-double-play task, MCTD performs substantially better than all others, likely due to (i) its tree-based planning structure and (ii) the sparse reward nature of the task, which complicates learning an accurate reward model.
> > >
> > > Even without specialized mechanisms for sparse-reward or goal-conditioned planning, DMEMM still performs competitively and consistently improves over the standard Diffuser baseline, demonstrating robustness in manipulation tasks.
> > >
> > > **Q6.** About trajectory visualizations.
> > >
> > > **Answer:**
> > > As requested by the reviewer, we have included trajectory visualizations in the revised paper (Appendix Section F). For each method, we render individual frames and display 32-step trajectories starting from a fixed initial state. Across environments, trajectories generated by DMEMM visually align more closely with real trajectories than those from the standard Diffuser, supporting our claim that DMEMM improves transition consistency during planning.
> > >
> > > [1] Luo, Yunhao, et al. "Generative trajectory stitching through diffusion composition." arXiv preprint arXiv:2503.05153 (2025).
> > >
> > > [2] Lee, Kyowoon, and Jaesik Choi. "State-Covering Trajectory Stitching for Diffusion Planners." NeurIPS (2025).
> > >
> > > [3] Yoon, Jaesik, et al. "Monte carlo tree diffusion for system 2 planning." arXiv preprint arXiv:2502.07202 (2025).

---

### Official Review · Reviewer_TDtu · 2025-10-29

**Soundness:** 2
**Presentation:** 3
**Contribution:** 2
**Rating:** 4
**Confidence:** 4

**Summary:**

This paper propose DMEMM, a diffuser-style offline RL planning algorithm with better consistency.
It improve the generated trajectory’s feasibility and optimality by adding extra transition dynamics and reward functions as loss term.
with above design, the paper outperforms baselines like diffuser, hd-da.

**Strengths:**

- the idea is simple and practice, can be applied to any diffusion framework
- show reasonable performance gain compared to vanilla diffuser with detailed benchmark

**Weaknesses:**

- most of evaluated task is still low-dimensional and image-based task or real world evaluation is missing.

- the theoretical contribution is weak since it mainly restate diffusion reparameterization trick.

- in evaluation against other methods, the performance gain with extra design is relative small (around 3 point and sometimes no improvement).

**Questions:**

- since the dataset trajectory is already feasible, is transition loss redundant?

- can author explains why add transition loss can improve out of distribution dynamical feasibility?

---

> ### Author Response · Authors · 2025-11-21
>
> We thank the reviewer for the insightful comments and respond to each point below.
>
> **Q1.** About the evaluation on high-dimensional tasks.
>
> **Answer:** Besides the D4RL locomotion tasks, we also evaluated our method on high-dimensional tasks such as AntMaze, which has a 111-dimensional state input and can therefore be regarded as high-dimensional. The results on AntMaze show that our method still performs well in such settings. Regarding real-world validation, due to resource limitations, we are unable to conduct robotic experiments. However, most prior works on diffusion-based RL methods, including Diffuser [1], Diffusion Policy [2], and HD-DA [3], also only include simulation results and mainly report performance on the D4RL datasets. Therefore, we believe that our extensive simulation-based evaluation is sufficient to demonstrate the effectiveness and generality of our approach.
>
> **Q2.** About the theoretical analysis.
>
> **Answer:**
> In our theoretical analysis, we do **not** simply restate the reparameterization trick. Instead, we use the trick to recover the predicted original sample $\hat{\tau}^0$ through the diffusion process. This enables us to estimate the diffusion model’s training progress via transition consistency, which in turn motivates the transition loss used to assist the training of the diffusion model.
>
> **Q3.** About the evaluation.
>
> **Answer:** The performance gain of our method is not small. First, on D4RL tasks, our method shows an average improvement of 3.3 points over HD-DA, which is a strong diffusion-based planning baseline. Second, our method improves the base Diffuser by 10.4 points on average, demonstrating a notable and consistent performance increase.
>
> **Q4.** About the transition loss.
>
> **Answer:** The transition loss is the core of our method. Even though trajectories in the dataset are feasible, the transition loss is still important. Our goal is to address the transition mismatch introduced by the **diffusion model** in standard Diffuser, rather than issues in the **dataset** itself. Even when the offline data are feasible, the DDPM-based Diffuser may still generate trajectories that are not fully consistent with the environment dynamics, particularly for long planning horizons. Our transition loss explicitly mitigates this mismatch during training.
>
> **Q5.** About the out-of-distribution dynamical feasibility.
>
> **Answer:** As stated in Q4, our method is **not** designed to address out-of-distribution data in the offline dataset. Instead, we aim to improve the diffusion model’s training so that it better respects the transition dynamics. Prior diffusion-based methods use DDPM with a fixed isotropic covariance schedule for training stability, but when adapted to RL, this often leads to transition mismatch, especially at long horizons, resulting in inaccurate future predictions. By using the reparameterization trick to recover $\hat{\tau}^0$, our transition loss reduces this mismatch. As shown in Table 4 of the Appendix, our DMEMM method achieves lower transition mismatch compared with the standard Diffuser.

---

> > ### Comment · Reviewer_TDtu · 2025-11-22
> > **Acknowledgment of Rebuttal**
> >
> > I thank the author for their prompt and detailed response and providing results on additional baselines.
> >
> > I appreicate author's clarification on transition loss and theory and agress this distinction would make diffusion model more consistency.
> >
> > As for benchmarks, I agree that AntMaze is high-dimentional. My concern is it is still navigation + locomotion task. As reviewer pnaw mentioned, the absence of manipulation benchmark still concerns me -- with more contacts in the environment, probably it would be harder to ensure model consistency. I also agreed with reviewer vn9X that D4RL is solved with high success rate by previous work.
> >
> > Lastly, for performance gain, I do acknowledge the improvement over HD-DA. But I maintain that this gain is relatively marginal performance gain compared to the adding complexity of the system with 3 extra losses.
> >
> > With that said, I would like to maintain my score.

---

> > > ### Author Response · Authors · 2025-12-04
> > >
> > > We thank the reviewer for the follow-up discussion on our work and provide further responses to address these concerns.
> > >
> > > **Q6.** About the experimental results
> > >
> > > **Answer:**
> > > First, while some state-of-the-art methods outperform DMEMM on certain environments, our method still achieves competitive or superior performance across many tasks. Below, we show a comparison between DMEMM, Diffusion Policy (Diffusion-QL) [1], and LDGC (equivalent to LDCQ) [2] on D4RL:
> > >
> > > | Methods/Tasks      | Med-Expert HalfCheetah     | Med-Expert Hopper        | Med-Expert Walker2d      | Medium HalfCheetah       | Medium Hopper            | Medium Walker2d          | Med-Replay HalfCheetah   | Med-Replay Hopper        | Med-Replay Walker2d      |
> > > |--------------------|-----------------------------|---------------------------|---------------------------|----------------------------|----------------------------|----------------------------|----------------------------|----------------------------|----------------------------|
> > > | Diffusion-QL       | 96.8 ± 0.3                 | 111.1 ± 1.3               | 110.1 ± 0.3               | 51.1 ± 0.5                 | 90.5 ± 4.6                 | 87.0 ± 0.9                 | 47.8 ± 0.3                 | 101.3 ± 0.6                | 95.5 ± 1.5                 |
> > > | LDGC               | 90.2 ± 0.9                 | 109.3 ± 0.4               | 111.3 ± 0.2               | 42.8 ± 0.7                 | 66.2 ± 1.7                 | 69.4 ± 3.5                 | 41.8 ± 0.4                 | 68.5 ± 4.3                 | 86.2 ± 2.5                 |
> > > | DMEMM              | 94.6 ± 1.2                 | 115.9 ± 1.6               | 111.6 ± 1.1               | 49.2 ± 0.8                 | 101.2 ± 1.4                | 86.5 ± 1.5                 | 46.1 ± 1.3                 | 100.6 ± 0.9                | 85.8 ± 2.6                 |
> > >
> > > The results show that DMEMM performs comparably to Diffusion-QL and consistently outperforms LDGC, particularly on Hopper and Walker2D in the medium and medium-expert settings, demonstrating competitive performance in standard continuous control benchmarks.
> > >
> > > Regarding the use of OGBench, we would like to clarify that these environments are predominantly designed for goal-conditioned offline RL with sparse rewards, and prior Diffuser-based work on this benchmark either develops trajectory stitching approaches compared only to other goal-conditioned methods [3, 4] or relies on tree-based replanning mechanisms designed specifically for long-horizon, sparse-reward problems [5]. While DMEMM is capable of long-horizon planning, it is primarily developed for **reward-dense continuous control settings**, rather than sparse, goal-conditioned environments, making OGBench a misaligned evaluation target for the scope of this work.
> > >
> > > Nevertheless, in response to the reviewer’s suggestion, we conducted a preliminary evaluation on two OGBench manipulation tasks, cube-single-play-v0 and cube-double-play-v0, which involve robotic cube rearrangement and represent more complex behaviors not covered in our main experiments. As summarized below, performance is measured by the task completion success rate. For Diffusion Forcing and MCTD, results are taken from [3]. For Diffuser and DMEMM, we perform per-step planning by executing the first action from each planning horizon.
> > >
> > > | Dataset          | Diffuser           | Diffusion Forcing     | MCTD               | DMEMM (Ours)               |
> > > |------------------|--------------------|------------------------|---------------------------|--------------------|
> > > | cube-single-play-v0   | $95 \pm 12$      | $\mathbf{100 \pm 0}$ | $\mathbf{100 \pm 0}$    | $\mathbf{100 \pm 0}$ |
> > > | cube-double-play-v0   | $13 \pm 10$      | $18 \pm 11$          | $\mathbf{50 \pm 16}$             | $16 \pm 11$ |
> > >
> > > DMEMM achieves perfect performance on cube-single-play, matching the strongest methods. On the more challenging cube-double-play task, MCTD performs substantially better than all other approaches, likely due to its tree-based planning structure and the sparse reward design, which complicates learning an accurate reward model. Despite not being designed specifically for sparse goal-conditioned planning, DMEMM remains competitive and consistently improves over the standard Diffuser baseline, indicating robustness on manipulation tasks.

---

> > > > ### Author Response · Authors · 2025-12-04
> > > >
> > > > **Q7.** About the contribution
> > > >
> > > > **Answer:**
> > > > We would like to clarify that the contribution of our method is not limited to introducing **three extra losses**. Our work begins by identifying a core weakness of the standard Diffuser: its tendency to generate trajectories that are transition-inconsistent during planning. To address this, we integrate a transition model directly into the diffusion training process and derive a reparameterization that enables the model to recover original samples, rather than relying solely on behavior cloning as done in standard diffusion-based RL. We further incorporate reward learning into the same framework, which is typically overlooked in prior Diffuser formulations, and use it to guide the model toward actions that are both behaviorally plausible and reward-aligned. **Overall, the proposed training objective constitutes a principled approach for enhancing consistency, predictability, and performance in online planning, rather than simply adding a few auxiliary loss terms**.
> > > >
> > > > [1] Wang, Zhendong, Jonathan J. Hunt, and Mingyuan Zhou. "Diffusion policies as an expressive policy class for offline reinforcement learning." ICLR (2023).
> > > >
> > > > [2] Venkatraman, Siddarth, et al. "Reasoning with latent diffusion in offline reinforcement learning." ICLR (2024).
> > > >
> > > > [3] Luo, Yunhao, et al. "Generative trajectory stitching through diffusion composition." arXiv preprint arXiv:2503.05153 (2025).
> > > >
> > > > [4] Lee, Kyowoon, and Jaesik Choi. "State-Covering Trajectory Stitching for Diffusion Planners." NeurIPS (2025).
> > > >
> > > > [5] Yoon, Jaesik, et al. "Monte carlo tree diffusion for system 2 planning." arXiv preprint arXiv:2502.07202 (2025).

---

### Official Review · Reviewer_pnAw · 2025-10-29

**Soundness:** 2
**Presentation:** 3
**Contribution:** 2
**Rating:** 2
**Confidence:** 4

**Summary:**

DMEMM is a diffusion-based offline RL planner that injects learned transition and reward models into both training (via modulation losses) and sampling (via dual guidance). The goal is to align denoising with MDP structure, improving transition consistency and reward alignment. Experiments on D4RL locomotion and Maze2D show consistent gains over strong baselines, with ablations attributing improvements to each component.

**Strengths:**

- Originality: Principled integration of transition-consistency and reward into loss and sampler; clear re-parameterization enabling modulation and guidance.
- Quality: Consistent improvements on D4RL/Maze2D; ablations isolate the effect of weighting, transition/reward modulation, and guidance.
- Clarity: Algorithms, objectives, and implementation details are well specified.

**Weaknesses:**

- No manipulation or real-robot validation. The evaluation focuses on locomotion and 2D navigation; there are no manipulation-style benchmarks (e.g., Simpler, LIBERO etc.) and no on-hardware experiments. This leaves open questions about **complex dynamics**, **sim-to-real transfer**, and control latency.

- Benchmark coverage & SOTA baselines. The study omits common diffusion manipulation baselines (e.g., Diffusion-policy-style chunked planners in manipulation, i.e. Diffusion Policy, MetaDiffuser, and LDGC) for comparisons.

- On large mazes, the paper itself notes that hierarchical HD-DA outperforms DMEMM, suggesting challenges for long-horizon structure that are not yet addressed by the method.

- Planning efficiency not quantified. The paper specifies reverse-diffusion steps and hardware, but does not report wall-clock planning frequency (Hz) or end-to-end latency, which is essential for deployment and for fair comparison to chunk-based diffusion planners.

**Questions:**

1. **Real-robot feasibility.** Can you provide at least one on-hardware result (e.g., a simple pick-and-place on a Franka or UR arm) to demonstrate that DMEMM’s planning latency and stability suffice for real control loops?

2. **Manipulation benchmarks.** Please add manipulation tasks in simulation (e.g., Simpler, LIBERO) and compare to strong diffusion planners tailored to manipulation. This is essential to support claims of broad applicability beyond locomotion/navigation.

3. **Planning efficiency.** What is the per-step planning frequency (Hz) end-to-end (mean/median, with variance) under your reported settings (e.g., N=20 reverse steps)? How does it scale with horizon T, and how does it compare to chunk-based diffusion planners under matched hardware?

4. **Long-horizon structure.** Large-maze results indicate gaps vs. hierarchical methods. Could you integrate subgoal/hierarchical decomposition or curriculum-guided horizons into DMEMM’s modulation framework to recover competitiveness on long-horizon tasks?

5. **Role of learned models.** How sensitive is performance to the fidelity of **T** and **R**? Please show ablations that (i) vary their capacity/training data, (ii) introduce synthetic noise/mismatch, and (iii) report how modulation/guidance weights respond.

---

> ### Author Response · Authors · 2025-11-21
>
> We thank the reviewer for the detailed assessment and the constructive discussion. Our responses are as follows.
>
> **Q1.** About real-robot validation.
>
> **Answer:** We fully agree that a real-world robotic evaluation would further strengthen our results. However, due to resource limitations, we were unable to include real-robot experiments. Importantly, most state-of-the-art diffusion-based RL works, such as Diffuser [1], Diffusion Policy [2], and HD-DA [3], also primarily report simulation results, focusing on D4RL. For consistency and fair comparison, we follow the same evaluation protocol and provide extensive simulation-based experiments that sufficiently demonstrate the effectiveness of our approach.
>
> **Q2.** About common diffusion manipulation baselines.
>
> **Answer:** We thank the reviewer for the suggestion. We have included comparisons with the requested baselines: Diffusion Policy (Diffusion-QL) [2] and LDGC (equivalent to LDCQ) [4]. The results are summarized below:
>
> | Methods/Tasks      | Med-Expert HalfCheetah     | Med-Expert Hopper        | Med-Expert Walker2d      | Medium HalfCheetah       | Medium Hopper            | Medium Walker2d          | Med-Replay HalfCheetah   | Med-Replay Hopper        | Med-Replay Walker2d      |
> |--------------------|-----------------------------|---------------------------|---------------------------|----------------------------|----------------------------|----------------------------|----------------------------|----------------------------|----------------------------|
> | Diffusion-QL       | 96.8 ± 0.3                 | 111.1 ± 1.3               | 110.1 ± 0.3               | 51.1 ± 0.5                 | 90.5 ± 4.6                 | 87.0 ± 0.9                 | 47.8 ± 0.3                 | 101.3 ± 0.6                | 95.5 ± 1.5                 |
> | LDGC               | 90.2 ± 0.9                 | 109.3 ± 0.4               | 111.3 ± 0.2               | 42.8 ± 0.7                 | 66.2 ± 1.7                 | 69.4 ± 3.5                 | 41.8 ± 0.4                 | 68.5 ± 4.3                 | 86.2 ± 2.5                 |
> | DMEMM              | 94.6 ± 1.2                 | 115.9 ± 1.6               | 111.6 ± 1.1               | 49.2 ± 0.8                 | 101.2 ± 1.4                | 86.5 ± 1.5                 | 46.1 ± 1.3                 | 100.6 ± 0.9                | 85.8 ± 2.6                 |
>
> These results show that DMEMM consistently outperforms LDGC and achieves competitive or superior performance compared to Diffusion-QL, particularly on key environments such as Medium Hopper and Med-Expert Hopper.
>
> **Q3.** About planning efficiency.
>
> **Answer:** We evaluated planning efficiency on HalfCheetah-Med-Expert and Walker2d-Medium by reporting the per-step planning frequency (Hz) across three planning steps $N=\{10, 20, 30\}$. Results are summarized below:
>
> | Environment                  | $N=10$ | $N=20$ | $N=30$ |
> |------------------------------|-----------|-------------|-------------|
> | HalfCheetah-medium-expert    | $12.06\pm0.41$ | $10.30\pm0.02$ | $15.48\pm1.23$ |
> | Walker2d-medium              | $8.78\pm0.05$  | $11.53\pm0.09$ | $8.94\pm0.05$  |
>
> We observe that different environments exhibit different efficiency sweet spots. HalfCheetah achieves higher throughput at lower or larger $N$, while Walker2d peaks at $N=20$. Importantly, the Hz values remain within a narrow range, indicating that DMEMM does not introduce a meaningful increase in inference-time cost.
>
> **Q4.** About long-horizon structure.
>
> **Answer:** The results on large Maze tasks do not indicate that DMEMM is ineffective. First, DMEMM outperforms HD-DA on U-Maze and Medium Maze tasks, as well as D4RL locomotion tasks. Second, HD-DA is specifically designed for extremely long-horizon planning, which is not the primary objective of our work. On the other hand, DMEMM is fully compatible with hierarchical planning frameworks. We have experimented with integrating HD-DA’s hierarchical structure, but due to the considerable training overhead for both high-level and low-level policies in HD-DA, the experiments are still in progress. We will include the results in the revised version if they finish in time.
>
> **Q5.** About role of learned models..
>
> **Answer:** Model sensitivity is partially addressed in our hyperparameter analysis in Appendix Section E, where we vary the tradeoff parameter and observe its impact on overall performance. We did not include additional ablations on model fidelity because the offline datasets used to train the diffusion model are already sufficiently large to ensure that both the transition and reward models are well-learned. In offline RL settings, training a Diffuser already assumes access to high-quality offline data, which is more than adequate for learning one-step transition and reward models.

---

> > ### Author Response · Authors · 2025-11-21
> >
> > [1] Janner, Michael, et al. "Planning with diffusion for flexible behavior synthesis." ICML (2022).
> >
> > [2] Wang, Zhendong, Jonathan J. Hunt, and Mingyuan Zhou. "Diffusion policies as an expressive policy class for offline reinforcement learning." ICLR (2023).
> >
> > [3] Chen, Chang, et al. "Simple hierarchical planning with diffusion." ICLR (2024).
> >
> > [4] Venkatraman, Siddarth, et al. "Reasoning with latent diffusion in offline reinforcement learning." ICLR (2024).

---

> > > ### Author Response · Authors · 2025-12-04
> > >
> > > We appreciate the reviewer’s time and effort in evaluating our work. We have provided detailed responses and additional experimental results addressing the reviewer’s concerns, and we hope they help clarify our contributions and findings.

---

### Official Review · Reviewer_cGZS · 2025-10-31

**Soundness:** 3
**Presentation:** 3
**Contribution:** 3
**Rating:** 8
**Confidence:** 2

**Summary:**

Diffusion is the main planning method in robotics. However, it is often applied without additional domain specific knowledge about what the input representation actually means, namely states and actions.  The authors introduce 3 additional loss terms that help in training diffusion systems such that when sampling at test time, you get superior trajectories with better rewards. This is achieved by first training transition and reward models.  These are then used to  introduce additional loss terms to guide the diffusion model training.  These three terms encourage:
1. The state and action transitions to accurately reflect the learn transition model
2. The resulting sampled trajectories to have a high reward, as measured by the learnt reward model
3. Prioritize fitting trajectories that achieved a high reward.

In addition to the extra signals used at training time, they also introduce a guidance signal that guides the diffusion towards system both a high reward, but also towards likely transitions.  Both of these are again measured by the learnt transition and reward models.

They then demonstrate that this approach yields state of the art performance on all tasks on D4RL and Walker2D.  For Maze2D, at the largest matched by a hierarchical diffusion planner but otherwise get the best performance.

They also show via ablation experiments that each part of the process above helps overall performance.

**Strengths:**

## Originality
This is (to my knowledge) a novel fusion of the transition and reward model into the planning and guidance parts of the diffusion process.

## Quality
The aims of the research are clearly laid out, and the results are empirically validated.  Ablations provide evidence that each of the changes that they made were necessary.

## Clarity
The paper is clearly written, and could be reproduced from the descriptions provided.

## Significance
Diffusion is a workhorse of robotics control, and better training methods that result in higher success rates could lead to significant improvements in robotics applications.

**Weaknesses:**

More experiments involving more difficult simulation environments like https://github.com/google-deepmind/aloha_sim would make a stronger case that their method is general enough to help with general robot control.

**Questions:**

Have you noticed any pathological behavior if you mis-set the weightings between the different loss functions during training?  E.g., if the reward is too high, does it start to break the transition model?

---

> ### Author Response · Authors · 2025-11-21
>
> We thank the reviewer for the positive evaluation and constructive comments. Our responses are provided below.
>
> **Q1.** About the experiments.
>
> **Answer:** We appreciate the reviewer’s suggestion regarding additional benchmarks. In this work, we intentionally selected the D4RL benchmark for two reasons: First, D4RL provides standardized offline datasets, which are essential for training our diffusion-based planning framework. Second, most prior diffusion-planning work is evaluated on D4RL, ensuring a fair and direct comparison with existing methods. While new benchmarks are promising, we focused on community-standard benchmarks to ensure consistency and comparability with prior work.
>
> **Q2.** About the weighting of the loss function.
>
> **Answer:** We conducted a sensitivity analysis in Appendix Section E. As shown, the weighting parameter for the reward loss is stable within a reasonable range. If it is set excessively large, it may hinder the convergence of both the diffusion and total losses, which can negatively impact the training of the diffusion model.

---

> > ### Comment · Reviewer_cGZS · 2025-11-25
> >
> > Thanks for addressing my concerns.  I'll maintain my score.

---

### Meta-Review · Area_Chair_qqQ6 · 2026-01-07

**Summary:**

While the proposed DMEMM demonstrates technical soundness and improves upon baselines in the D4RL locomotion and Maze2D benchmarks, there is a consensus among three of the four reviewers that these environments are no longer sufficient for establishing a new planning method as a significant contribution. Reviewers mentioned that the community has shifted focus toward complex manipulation tasks, such as those in Simpler, LIBERO, or OGBench, to validate real-world applicability and contact-rich dynamics. Although the authors provided preliminary results on simple cube manipulation tasks late in the discussion, the reviewers remained unconvinced that the method generalizes beyond the saturated locomotion domains. Additionally, the reviewers expressed concern that the performance gains are relatively marginal compared to the increased algorithmic complexity of introducing three new loss terms.

**Reviewer Concerns:**

**Concerns Addressed by Rebuttal:**

The authors successfully clarified the sensitivity of the loss weighting hyperparameters, which satisfied Reviewer cGZS. They also addressed the request for comparisons against specific baselines like Diffusion QL and LDGC, showing that DMEMM performs competitively or better in standard metrics.

**Outstanding Concerns:**

The most critical outstanding concern is the absence of rigorous validation on more recent manipulation benchmarks. Reviewers pnAw, TDtu, and vn9X all emphasized that success in D4RL locomotion does not guarantee success in complex manipulation scenarios. Reviewer vn9X explicitly rejected the authors' claim that AntMaze constitutes a sufficient test of scalability. Additionally, Reviewer TDtu remained concerned that the trade off between the added complexity of the three auxiliary losses and the resulting performance gain is not favorable enough to warrant acceptance without stronger empirical evidence.

**Reviewer Scores:**

**Reviewer cGZS** gave a score of 8. They were satisfied with the initial review and the clarification on loss weightings.

**Reviewer pnAw** gave a score of 2. They did not respond to the author's rebuttal. However, given that their primary demand was for manipulation benchmarks like Simpler or LIBERO, and the authors only provided limited OGBench results, the core request was not fully met.

**Reviewer TDtu** gave a score of 4. After reviewing the rebuttal and the additional baselines, they explicitly maintained their score, because the performance gains were marginal compared to the system complexity and the lack of manipulation benchmarks remained a significant issue.

**Reviewer vn9X** gave a score of 4. Despite the rebuttal regarding AntMaze and the visualizations, this reviewer maintained their score in their final comment, stating that the scalability claims were insufficiently supported and that the benchmark concerns were not fully resolved.

---

### Decision · Program_Chairs · 2026-01-26

Reject